# Augmentation-Driven Metric for Balancing Preservation and Modification in Text-Guided Image Editing

## Abstract

The development of vision-language and generative models has significantly advanced text-guided image editing, which seeks *preservation* of core elements in the source image while implementing *modifications* based on the target text. However, in the absence of evaluation metrics specifically tailored for text-guided image editing, existing metrics are limited in their ability to balance the consideration of both preservation and modification. Especially, our analysis reveals that CLIPScore, the most commonly used metric, tends to favor modification, resulting in inaccurate evaluations. To address this problem, we propose `AugCLIP`, a simple yet effective evaluation metric that balances preservation and modification. `AugCLIP` begins by leveraging a multi-modal large language model (MLLM) to augment detailed descriptions that encapsulate visual attributes from the source image and the target text, enabling the incorporation of richer information. Then, `AugCLIP` estimates the modification vector that transforms the source image to align with the target text with minimum alteration as a projection into the hyperplane that separates the source and target attributes. Additionally, we account for the relative importance of each attribute considering the interdependent relationships among visual attributes. Our extensive experiments on five benchmark datasets, encompassing a diverse range of editing scenarios, demonstrate that `AugCLIP` aligns remarkably well with human evaluation standards compared to existing metrics. The code for evaluation will be open-sourced to contribute to the community.

## 1 Introduction

Building on advancements in vision-language models (Radford et al., 2021; Li et al., 2022; Geng et al., 2023), recent generative models (Kawar et al., 2022; Brooks et al., 2022; Hertz et al., 2022) have been widely utilized as creative tools for image editing via text instructions. Text-guided image editing models enable the modification of images in response to textual guidance, ensuring that changes are aligned with the provided instructions. The primary objective of these models is to apply specific *modifications* guided by the target text while *preserving* the core attributes of the source image.

Despite the remarkable advancements in editing models, there has been a lack of rigorous evaluation methods, tailored specifically for text-guided image editing. Consequently, most studies (Hertz et al., 2023; Basu et al., 2023; Gal et al., 2022; Kim & Ye, 2021; Brooks et al., 2022; Gal et al., 2022; Ruiz et al., 2023; Kocasari et al., 2022) have heavily relied on human evaluation, which provides balanced consideration of preservation and modification aspects. However, as it is costly and impractical for real-world applications, researchers have adapted automatic evaluation metrics (Zhang et al., 2018; Kim & Ye, 2021; Caron et al., 2021; Gal et al., 2022) originally designed for other vision tasks, such as image generation or captioning. Particularly, CLIPScore (Gal et al., 2022) is widely used as a representative metric, which evaluates the extent of alignment between the edited image and the target text, based on the difference between the target and source text in the CLIP space.

However, despite its widespread adoption, our analysis reveals significant limitations in CLIPScore, contradicting the standard of human evaluators. First, it tends to prioritize modification over preservation, unlike human evaluators who balance both aspects. This bias leads to inflated scores for excessively modified images that neglect even key attributes of the source image. Second, CLIPScore often focuses on peripheral parts rather than regions that are pertinent to the target text, whereas human evaluators can focus on the regions that must be edited. These observations underscore the need to reevaluate the effectiveness of CLIPScore in text-guided image editing.

Based on our comprehensive analysis, we propose a novel metric, `AugCLIP`, which evaluates the quality of edited images by comparing with an estimated representation of a well-edited image that balances preservation and modification by identifying a key modification vector that transforms the source image to match the target text while minimizing alterations. For this purpose, we leverage large language models to extract attributes that capture various visual aspects of the source image and target text. Then, we estimate the key modification vector by a hyperplane that separates the source and target attributes, considering the intertwined relationships between them. To this end, `AugCLIP` evaluates how closely the edited image aligns with the estimated ideal derived by applying the modification vector to the source image.

Our metric `AugCLIP` demonstrates remarkable improvement in alignment with human evaluators on diverse editing scenarios such as object, attribute, style alteration compared to all other existing metrics. Moreover, our metric is even applicable to personalized generation, DreamBooth dataset, where the objective is to identify the source object in provided image, and generate into a completely novel context. This shows the flexibility of `AugCLIP`, that seamlessly apply to variety of editing directions. Notably, our metric excels in identifying minor differences between the source image and the edited image, showing superb ability in complex image editing scenarios such as MagicBrush.

The major contributions are summarized as follows.

- We are the first to point out CLIPScore's reliability in text-guided image editing, as it frequently exhibits a bias towards modification rather than preservation and focuses on irrelevant regions.

- We introduce `AugCLIP`, a metric for image editing by automatically augmenting descriptions via LLM and estimating a balanced representation of preservation and modification, which takes into account the relative importance of each description.

- `AugCLIP` demonstrates a significantly high correlation with human evaluations across various editing scenarios, even in complex applications where existing metrics struggle.

## 2 Related Works

Currently widely used metrics for text-guided image editing assess one of the following aspects: image quality and image-text alignment. For evaluating image quality, FID (Heusel et al., 2017), IS (Salimans et al., 2016), and LPIPS (Zhang et al., 2018) measure feature distance between generated images and real images. Additionally, DiffusionCLIP (Kim & Ye, 2021) introduces a disentanglement metric called segmentation consistency, which compares segmentation maps of source and edited images under the assumption that the shape remains unchanged. However, these metrics tend to focus primarily on the preservation of the source image rather than assessing the quality of the modifications. To evaluate image-text alignment, CLIPScore (Gal et al., 2022) is widely used, measuring the similarity between the intended textual change and the actual modifications in the image, helping to assess how well the source image is altered according to the target text.

Several works explore image generation or image fidelity evaluation with CLIP-based metrics (Jayasumana et al., 2024; Kirstain et al., 2023; Kim et al., 2023; Lu et al., 2024). Li et al. (2024) bears similarity to our approach, particularly in its use of Large Language Models (LLMs) to extract detailed aspects. Nonetheless, this work focuses on image generation,

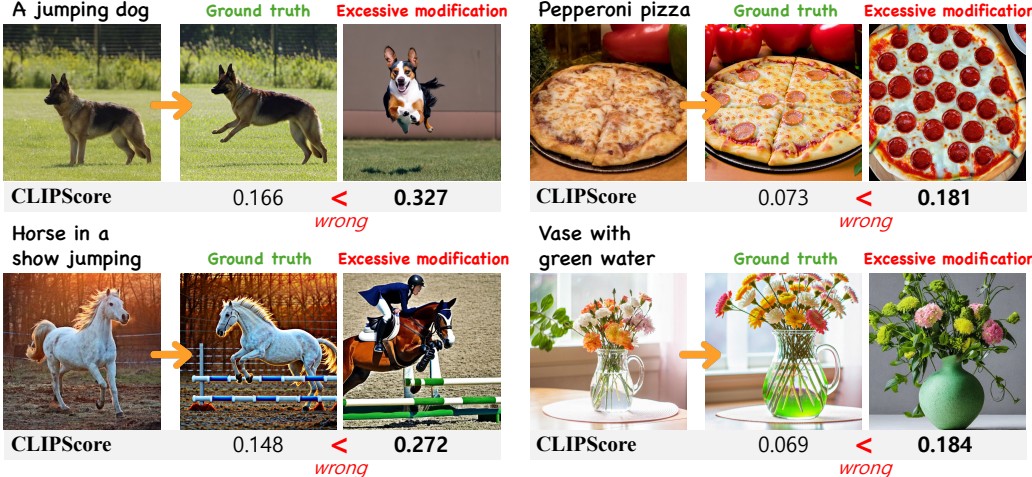

**Figure 1: CLIPScore's Bias towards Modification over Preservation**. Examples of cases in the TEdBench dataset, where CLIPScore assigns higher scores on excessively modified images over well-edited ground truth images. Similar observations persist over many cases in the TEdBench and MagicBrush datasets, where modification bias prevails over source image preservation. The samples used in the experiment are provided in the appendix.

making it less suited for editing tasks, where the preservation of original content alongside modifications is critical.

In contrast, our proposed metric, `AugCLIP`, provides a comprehensive evaluation that accounts for both preservation and modification. This dual assessment ensures that models make appropriate changes while retaining essential features of the source image, offering a more nuanced evaluation than existing metrics.

## 3    PROBLEM ANALYSIS ON EXISTING METRICS FOR TEXT-GUIDED IMAGE EDITING MODEL

In this section, we discover two major challenges in CLIPScore as an evaluation metric for text-guided image editing. First, CLIPScore tends to overemphasize modification aligning with the target text while neglecting the preservation of the source image (Sec. 3.2). Second, it often fails to concentrate on the image regions that are directly relevant to the target text (Sec. 3.3).

### 3.1    PRELIMINARIES: CLIPSCORE

In common text-guided image editing scenarios, a model generates an edited image $I_{\text{edit}}$ from a source image $I_{\text{src}}$ accompanied by a target text $T_{\text{trg}}$. Additionally, a source text $T_{\text{src}}$ that represents the source image is either provided as descriptions annotated by humans or generated using image captioning models.

CLIPScore, the most widely used metric in text-guided image editing, evaluates the modification based on the difference between $T_{\text{trg}}$ and $T_{\text{src}}$ in the CLIP space as follows:

$$\text{CLIPScore} = \text{cs}(\Delta I, \Delta T) = \text{cs}\Big(\text{CLIP}(I_{\text{edit}}) - \text{CLIP}(I_{\text{src}}), \text{CLIP}(T_{\text{trg}}) - \text{CLIP}(T_{\text{src}})\Big), \quad (1)$$

where $\text{cs}(\mathbf{a}, \mathbf{b}) = \frac{\mathbf{a} \cdot \mathbf{b}}{\|\mathbf{a}\|\|\mathbf{b}\|}$ denotes cosine similarity and $\text{CLIP}(\cdot)$ is a CLIP encoder for either image or text.

### 3.2    OVEREMPHASIZING MODIFICATION OVER PRESERVATION IN EVALUATION

Although CLIPScore attempts to incorporate the preservation by subtracting $T_{\text{src}}$ from $T_{\text{trg}}$, we observe that it has a tendency to overemphasize modifications towards target text. In

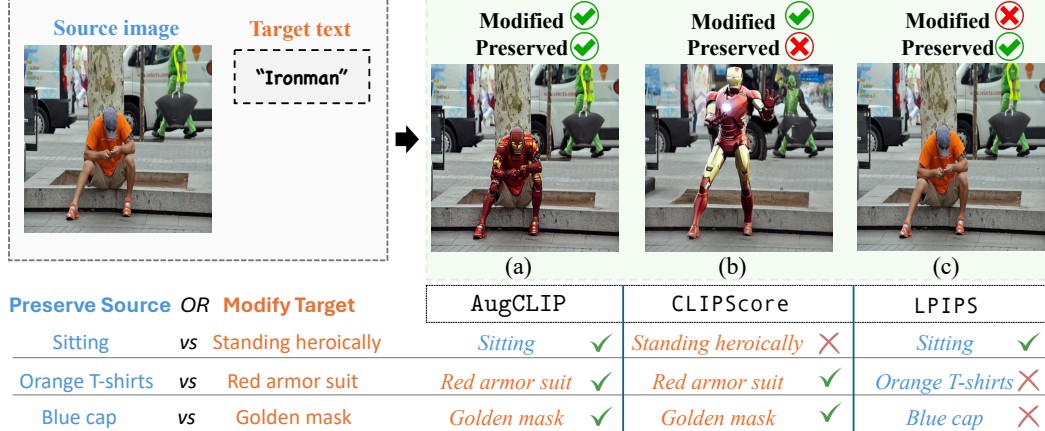

**Figure 2: Problem Setting for Evaluation in Text-Guided Image Editing.** An example of evaluation metrics for assessing edited images, with an aim to balance modifications relevant to the target text while preserving key aspects of the original image. (a) `AugCLIP` correctly determine retention or modification case-by-case. (b) CLIPScore incorrectly emphasizes excessive modification, preferring a standing Ironman. (c) LPIPS focuses on preservation, failing to apply necessary modifications, such as the Ironman suit. This demonstrates the need for an evaluation metric that judiciously balances both modification and preservation to achieve harmonious edits.

Fig. 1, CLIPScore often assigns higher scores to excessively modified images that neglect the key aspects of the source image.

To investigate this further, we conduct an experiment on the TEdBench (Kawar et al., 2022) and MagicBrush (Zhang et al., 2024) datasets, which consist of pairs of source images and target texts, along with ground truth edited images reflecting the desired edits. We generate excessively modified images using the text-to-image generation model, Stable Diffusion 1.5, based solely on the target text. Our results show that CLIPScore struggles to differentiate between ground truth images and excessively modified ones, favoring ground truth images in only 37% of cases in Tedbench, and 64.9% of cases in MagicBrush. This highlights CLIPScore's bias toward modification over preservation.

This inability of CLIPScore to properly account for the source image preservation stems from its design of the text direction, which assumes that a well-edited image should primarily adhere to the target text. As illustrated in Fig. 2, conflicts frequently occur between the visual elements of the source image and the target text regarding which features should be preserved or modified. For example, the 'sitting' posture of the source image should be preserved over the 'standing heroically' description in the target text, while the 'orange T-shirt' should be modified to a 'red armor suit.' A well-designed metric would account for these conflicts, but CLIPScore, due to its underlying assumption, blindly favors features from the target text, leading to unreliable results. This highlights the need for a metric that better balances preservation and modification.

### 3.3 Overlooking Edited Regions in the Image

An evaluation metric is more effective when it focuses on the image regions modified following the target text, rather than peripheral or unchanged regions. For example, if a target text specifies making a dog yawn, the evaluation metric works better when it concentrates primarily on the dog's mouth, not its ears. To assess CLIPScore's capability in this regard, we conduct an experiment using the relevancy map (Chefer et al., 2021), denoted as $\boldsymbol{R}$, which visualizes the transformer's attention on an image corresponding to a given text. Specifically, for an image $I \in \mathbb{R}^{h \times w}$ and text $T$, the relevancy map is computed as $\boldsymbol{R}(I;T) = \nabla_{\boldsymbol{A}} \mathrm{cs}(\mathrm{CLIP}(I), \mathrm{CLIP}(T); \boldsymbol{A}) \odot \boldsymbol{A} \in \mathbb{R}^{h \times w}$, where $\boldsymbol{A}$ represents the attention scores of the CLIP visual encoder and $\odot$ denotes the Hadamard product. To visualize the relevancy map of CLIPScore, which is a cosine similarity between $\Delta I$ and $\Delta T$, we subtract the two relevancy maps as $\boldsymbol{R}(\Delta I; \Delta T) = \boldsymbol{R}(I_{\mathrm{edit}}; \Delta T) - \boldsymbol{R}(I_{\mathrm{src}}; \Delta T)$.

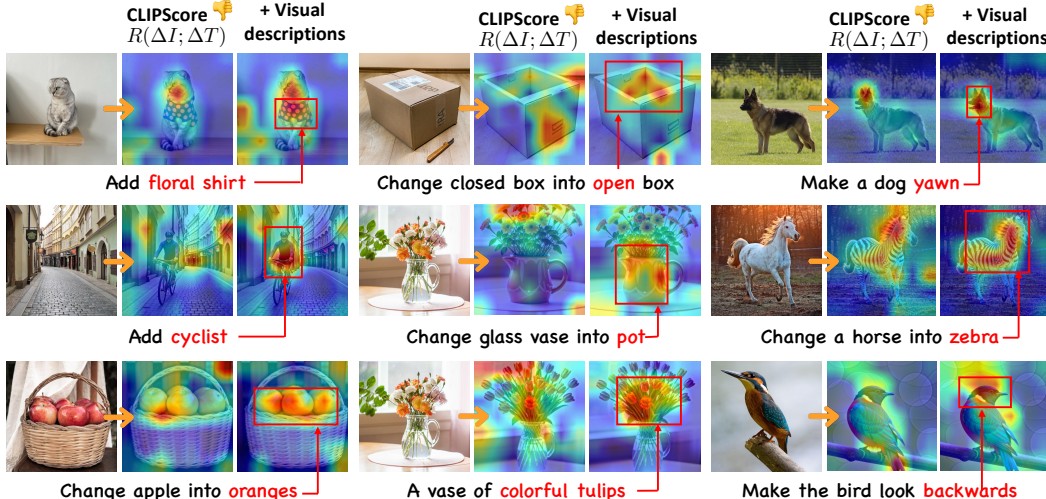

**Figure 3: CLIPScore's Overlooking Edited Regions in the Image**. This figure compares the relevancy map of CLIPScore $R(\Delta I; \Delta T)$ with the description-augmented relevancy map across various editing scenarios. The text related to the edits is written in red. CLIPScore highlights regions of the edited images irrelevant to the target text. However, when manually annotated visual descriptions are added, the relevancy maps demonstrate a significant improvement in accurately localizing the edited regions in red boxes as indicated by the target text.

Fig. 3 illustrates the relevancy maps of CLIPScore, $R(\Delta I; \Delta T)$, for randomly selected text-image pairs in TEdBench. We observe that CLIPScore is not an ideal metric as it often fails to attend to image regions relevant to the target text. This limitation arises because the target text alone does not fully capture the detailed aspects of desired edits. To provide the missing details, we use manually annotated visual descriptions, such as 'opened mouth' and 'pink tongue extended' for the target text 'yawn'. As shown in Fig. 3, this additional information enables CLIPScore to more accurately attend to the relevant regions as demonstrated in Fig. 3. This suggests that more explicit descriptions of essential attributes can improve the effectiveness of image editing evaluations.

## 4 AugCLIP: A Novel Metric Balancing Preservation and Modification

In this section, we propose a novel evaluation metric, `AugCLIP`, that estimates the representation of a well-edited image by identifying a key modification vector that transforms the source image to match the target text while minimizing alterations. `AugCLIP` starts by augmenting the source image and target text with fine-grained attributes (Sec. 4.1). Then, the key modification vector is determined by identifying the normal vector of a hyperplane that separates the source and target attributes, balancing the preservation of the source image with the modifications required by the target text (Sec. 4.2). In this process, we also account for the relative importance of each visual attribute, considering their interrelationships in response to the target text (Sec. 4.3).

### 4.1 Extracting Visual Attributes

Inspired by the finding that detailed descriptions of the target text make the edited region more noticeable in Sec. 3.3, we extract visual attributes from the source image and target text using a state-of-the-art multi-modal large language model (MLLM), GPT-4V (OpenAI, 2023). To extract visual attributes from the source image, we prompt GPT-4V to generate a detailed caption that encapsulates the key visual attributes present in the source image. This caption is then parsed into discrete visual attributes. For example, given a source image depicted in Fig. 2, let us assume that GPT-4V generates the caption: 'a man is sitting and wearing both blue caps and orange T-shirt'. Then, this caption is broken down into individual attributes such as 'a sitting man', 'wearing a T-shirt', and 'wearing a blue cap.'

When processing the target text, the focus shifts to identifying the modifications that need to be made to the source image during the editing process. To achieve this, GPT-4V is prompted with both the source and target text and then instructed to describe the aspects of the target text that diverge from the source text. To ensure that each generated description corresponds to a single visual attribute, we provide example descriptions along with the prompt.

These attributes are encoded into CLIP, where the source attributes are denoted as $\boldsymbol{S} = \{\mathbf{s}_i\}_{i=1}^{N_s}$ and target attributes as $\boldsymbol{T} = \{\mathbf{t}_j\}_{j=1}^{N_t}$. $N_s$ and $N_t$ are the number of attributes for the source and target, respectively. The detailed prompting process and statistics are described in the appendix.

## 4.2 DERIVING THE KEY MODIFICATION VECTOR

Based on the source and target attributes extracted in Sec. 4.1, we identify a key modification vector $\mathbf{v}$ in CLIP space, representing the *minimum modification* that adjusts the source image to align with the target text. Essentially, $I_{\text{src}} + \mathbf{v}$ approximates a well-edited image.

Here, the direction of $\mathbf{v}$ is chosen to highlight the differences between the source image and the target text. An intuitive way to estimate such direction is by using the normal vector $\mathbf{w}$ of the decision boundary that separates the source distribution, $\boldsymbol{S}$, from the target distribution, $\boldsymbol{T}$. Formally, the classifier function $f(x) = \mathbf{w}^T x + b$ assigns $x$ to the 'target' class if $f(x) > 0$, or to the 'source' class if $f(x) < 0$. Since this classification relies on the projection onto the normal vector $\mathbf{w}$, this vector captures the attributes most distinguishing between $\boldsymbol{S}$ and $\boldsymbol{T}$.

Then, the $\mathbf{v}$ is a vector that has minimum norm and satisfies the condition that the edited image is classified as belonging to the 'target' class ($f(I_{\text{src}} + \mathbf{v}) > 0$):

$$\min_{\mathbf{v}} \quad \|\mathbf{v}\| \quad \text{subject to} \quad \mathbf{w}^T \mathbf{v} > -(\mathbf{w}^T I_{\text{src}} + b). \tag{2}$$

Finally, the modification vector is expressed as

$$\mathbf{v} = \frac{\mathbf{w}^\top I_{\text{src}} + b}{\|\mathbf{w}\|^2} \mathbf{w}, \tag{3}$$

which represents the projection of the source image $I_{\text{src}}$ onto the decision boundary.

## 4.3 CONSIDERING INTERTWINED RELATIONSHIP BETWEEN ATTRIBUTES

When determining the separating hyperplane between the source and target attributes, it is crucial to account for the relative importance of each attribute, considering the interconnections between them. This is because most image editing tasks require simultaneous modification of multiple related visual attributes, as these attributes often work together to create a cohesive appearance. For instance, transforming a human face into a 'smiling face' involves adjusting several interconnected features, such as upturned mouth corners, crinkled eyes, and raised cheeks, all of which must appear together in the edited image. However, the current approach to defining the hyperplane focuses solely on separation and does not consider these attribute relationships.

To address this, we refine the hyperplane optimization process so that $\mathbf{v}$ reflects the interdependencies between attributes. Specifically, we enhance the cohesiveness of attributes within the same class to quantify their degree of interrelation, and use this information to weigh each attribute during optimization. Additionally, source or target attributes that are already similar to those in the opposite class (*i.e.*, target or source, respectively) are less relevant to the editing process and thus have less impact on the modification vector. As a result, their influence should be reduced during the hyperplane optimization.

The final weightings, $\boldsymbol{a}_s$ for source attributes and $\boldsymbol{a}_t$ for target attributes, are then defined as:

$$\boldsymbol{a}_s^{(i)} = \mathbb{E}_{\boldsymbol{s} \in \boldsymbol{S}}[\mathbf{cs}(\boldsymbol{s}_i, \boldsymbol{s})] - \mathbb{E}_{\boldsymbol{t} \in \boldsymbol{T}}[\mathbf{cs}(\boldsymbol{s}_i, \boldsymbol{t})] \qquad \text{for } \boldsymbol{s}_i \in \boldsymbol{S}, \tag{4}$$

$$\boldsymbol{a}_t^{(j)} = \mathbb{E}_{\boldsymbol{t} \in \boldsymbol{T}}[\mathbf{cs}(\boldsymbol{t}_i, \boldsymbol{t})] - \mathbb{E}_{\boldsymbol{s} \in \boldsymbol{S}}[\mathbf{cs}(\boldsymbol{t}_i, \boldsymbol{s})] \qquad \text{for } \boldsymbol{t}_j \in \boldsymbol{T}. \tag{5}$$

Then, the refined version of $\mathbf{v}$ is obtained through hyperplane optimization using $\boldsymbol{a}_s$ and $\boldsymbol{a}_t$.

Finally, `AugCLIP` evaluates how the edited image aligns with the estimation of the well-edited image in CLIP space as

$$\texttt{AugCLIP} = \texttt{cs}(I_{\text{edit}}, I_{\text{src}} + \mathbf{v}), \tag{6}$$

where $\mathbf{v} = (\mathbf{w}^\top I_{\text{src}} + b)/\|\mathbf{w}\|^2 \cdot \mathbf{w}$ from Eq. (3).

## 5 EXPERIMENTS

**Implementation details.** For our experiments, we employ a pre-trained CLIP-ViT 16/B model for CLIP-based metrics. Source and target attributes are generated using GPT-4V (OpenAI, 2023). Further details on prompting the source and target descriptions are deferred to the appendix due to spatial constraints.

**Compared Metrics.** We compare `AugCLIP` with two categories of existing metrics. The first category comprises the metrics that focus solely on preservation aspects, including DINO similarity, LPIPS, and L2 distance. The other category measures target text alignment, for which the only metric is CLIPScore. Additionally, we utilize description-augmented versions of CLIPScore.

**Evaluation Datasets.** We evaluate `AugCLIP` and existing metrics across several text-guided image editing benchmarks, including TEdBench (Kawar et al., 2022), EditVal (Basu et al., 2023), MagicBrush (Zhang et al., 2024), DreamBooth (Ruiz et al., 2023), and CelebA Liu et al. (2015).

**Table 1: Difference Types of Benchmark Datsaet in Text-guided Image Editing**

|  | CelebA | EditVal | DreamBooth | TEdBench | MagicBrush |
|---|---|---|---|---|---|
| **Dataset Types** | Facial Attribute | General Object | Personalized Generation | Object Centric | Local Region Editing |

### 5.1 QUALITY ASSESSMENT ON EVALUATION METRICS

To evaluate the effectiveness of different evaluation metrics, we conduct two types of experiments, named 2AFC test and Ground truth test. **Two-Alternative Forced Choice (2AFC) test** (Tab. 2a) reveals the alignment between the evaluation score and human judgment. In this test, human evaluators are asked with two options of edited images, and then to choose the one they favor through the systematic survey. The alignment score measures if the evaluation metric prefers the same option as human evaluators. Secondly, **Ground Truth Test** (Tab. 2b) assess the ability of evaluation metric to correctly assign the highest score to the well-edited image among a triplet of images, (well-edited, excessively modified, excessively preserved). Yielding high scores in this test means that the evaluation metric can balancedly consider preservation and modification aspect, without being biased to either side.

**Two-Alternative Forced Choice (2AFC) test** The 2AFC score, denoted as $\boldsymbol{s}_{\text{align}}$, ranges from 0 to 1, where 1 indicates perfect alignment between an evaluation metric and human judgment. Tab. 2a demonstrates a comparison between `AugCLIP` and other evaluation metrics, across three benchmark datasets: CelebA, EditVal, and DreamBooth. These three text-guided image editing benchmark dataset represent a very distinct editing scenario. First, CelebA focuses on fine-grained editing of facial attributes such as eyebrows or lips. EditVal is a dataset that consists of general object modification, oftentimes including multiple objects in the source image. The target text instructions guide various types of editing such as style transfer, size transformation, and attribute alteration. Finally, Dream-Booth is a dataset tailored for personalized text-guided image generation, which aims to preserve the identity of the object depicted in the image while generating in a completely different contextual background.

Among these three datasets of different scenarios, CLIPScore demonstrates the competitive level of alignment with human judgment, as $\boldsymbol{s}_{\text{align}}$ scored 0.673 and 0.697 for CelebA and

**Table 2: Comparison on `AugCLIP` and Other Existing Metrics.** (a) 2AFC Test. The alignment score $s_{\text{align}}$ between human judgment and the evaluation metric is compared over three datasets, CelebA, EditVal, and Dreambooth. (b) Ground Truth Test. The accuracy of assigning higher scores to ground truth images over excessively preserved and modified images ($\textbf{Acc}_{\text{both}}$) are compared on two datasets, TEdBench and MagicBrush.

**\* DINO**: DINO similarity, P, M: consideration of preservation and modification, the best results are emphasized in **bold** font and the second best in underline.

|  | P | M | (a) 2AFC Test | | | (b) Ground Truth Test | |
|---|---|---|---|---|---|---|---|
|  |  |  | CelebA | EditVal | DreamBooth | TEdBench | MagicBrush |
|  |  |  | $s_{\text{align}}$ | $s_{\text{align}}$ | $s_{\text{align}}$ | $\textbf{Acc}_{\text{both}}$ | $\textbf{Acc}_{\text{both}}$ |
| **L2** | ✓ | ✗ | 0.653 | 0.348 | 0.464 | 0.310 | 0.002 |
| **LPIPS** | ✓ | ✗ | 0.465 | 0.360 | 0.286 | 0.090 | 0.000 |
| **DINO** | ✓ | ✗ | 0.574 | 0.348 | 0.286 | 0.280 | 0.008 |
| **CLIPScore** | ▲ | ✓ | 0.673 | 0.697 | 0.357 | 0.350 | 0.601 |
| `AugCLIP` | ✓ | ✓ | **0.883** | **0.831** | **0.857** | **0.570** | **0.889** |

EditVal, respectively. However, in the specific setting of personalized text-guided image generation, CLIPScore largely fails to align with human judgments, scoring merely 0.357. `AugCLIP`, which is augmented by rich visual semantics to flexibly be adapted into a difficult editing scenario, shows remarkable improvement in alignment score from 0.357 to 0.857.

**Ground Truth Test**  Among the triplet of three images, (well-edited image, excessively preserved, and excessively modified), the well-edited image is provided in the benchmark dataset, TEdBench, and MagicBrush. Excessively preserved images are generated by applying noise jitter on the source image, completely disregarding the target text. Excessively modified images are generated using the text-to-image generation model, Stable Diffusion 1.5, to generate the image instructed by the target text, while completely ignoring the source image.

Given the triplet of three images, we count the number of cases where evaluation metrics correctly assign the highest score to the well-edited image, and denote this count over all test cases in the benchmark dataset as $\textbf{Acc}_{\text{Both}}$. High accuracy reflects a metric's ability to balance both preservation and modification. In Tab. 2b, we observe that CLIPScore has a low $\textbf{Acc}_{\text{both}}$ score, failing on 65 % of the cases in TEdBench triplets, and on 39.9% of MagicBrush triplets. This observation corresponds to the problem analysis in Sec. 3.2, which pointed out the problem of CLIPScore favoring excessive modification, even ignoring preservation aspects. This proves that CLIPScore falls short of balancing the source preservation and target modification aspects. Such inability is also observed by In contrast, `AugCLIP`, which balances preservation and modification aspects through the estimation of an ideal image with separating hyperplane, scores the highest accuracy among all datasets and baseline metrics.

## 5.2 Ablation Study

In this section, we conduct an ablation of the effect of integrating visual attributes into CLIPScore. The original formulation of CLIPScore is simply a subtraction between the target text and source text. We experiment with the strategy of simply augmenting CLIP-

**Table 3:** Effect of Augmenting Descriptions into CLIPScore

|  | CelebA | EditVal | DreamBooth | TEdBench | MagicBrush |
|---|---|---|---|---|---|
| **CLIPScore** | 0.673 | 0.697 | 0.357 | 0.350 | 0.601 |
| + src desc. | 0.816 | 0.629 | 0.357 | 0.400 | 0.429 |
| + trg desc. | 0.819 | 0.708 | 0.536 | 0.420 | 0.533 |
| + both | 0.816 | 0.607 | 0.536 | 0.440 | 0.402 |
| `AugCLIP` | **0.883** | **0.831** | **0.857** | **0.570** | **0.889** |

Score with source and target descriptions, extracted in Sec. 4.1. First, the source text is augmented by averaging all CLIP features of the source descriptions. In Tab. 3, '+src desc' shows the effect of this strategy. In CelebA, source augmentation has led to a gain in alignment score, but in the other four datasets, the performance rather dropped. Second, the target text is augmented by averaging the CLIP features of target descriptions. This has led to a small gain in performance in CelebA, EditVal, DreamBooth, and TEdBench. However, MagicBrush suffers from a drop in accuracy. Finally, the third variant is to augment both source text and target text with corresponding descriptions. This strategy fails to improve over the second strategy, augmenting the target text only, except for the dataset TEdBench.

`AugCLIP` outperforms all the description-augmented variants of CLIPScore. The major difference between these simple strategy and `AugCLIP` is firstly a absence of weighting strategy that captures the relative importance of each attribute. Moreover, `AugCLIP` derives a minimum modification vector in the form of projection into separating hyperplane between source and target descriptions. CLIPScore is a simple method that subtracts the difference between source and target. This suggests that our approach, which estimates a well-edited image by discovering only necessary attributes, and inflicting only the necessary modification is meaningful, as mere description augmentation does not provide improvement in most of the cases.

**Table 4: Ablation study.** We use $s_{\text{align}}$ for CelebA, EditVal, and Dreambooth, and $\textbf{Acc}_{\text{both}}$ for TEdBench and MagicBrush. `AugCLIP` extracts short descriptions with unrestricted numbers.

**(a)** Effect of weighting strategy.

|  | CelebA | EditVal | DreamBooth | TEdBench | MagicBrush |
|---|---|---|---|---|---|
| Unweighted | 0.849 | 0.787 | 0.786 | 0.400 | 0.830 |
| Weighted | **0.883** | **0.831** | **0.857** | **0.570** | **0.889** |

**(b)** Effect of choice of linear hyperplane.
* Average misclassification rate of source attributes and target attributes in hyperplane fitting.

|  | CelebA | EditVal | DreamBooth | TEdBench | MagicBrush | Average Misc.* |
|---|---|---|---|---|---|---|
| LDA | **0.884** | 0.827 | 0.821 | 0.545 | 0.863 | 0.0337 |
| Logistic regression | 0.849 | 0.830 | 0.821 | 0.550 | 0.866 | 0.0138 |
| Linear SVM | 0.883 | **0.831** | **0.857** | **0.570** | **0.889** | **0.0135** |

**(c)** Effect of length and number of descriptions.

| Length | Number | CelebA | EditVal | DreamBooth | TEdBench | MagicBrush |
|---|---|---|---|---|---|---|
| short | 10 | **0.870** | 0.719 | **0.857** | 0.540 | **0.889** |
| short | 20 | 0.829 | **0.809** | 0.821 | 0.540 | 0.868 |
| short | 30 | 0.829 | 0.764 | 0.714 | **0.570** | 0.863 |
| long | 30 | 0.843 | 0.697 | 0.750 | 0.530 | 0.862 |
| short | unrestricted | **0.883** | **0.831** | **0.857** | **0.570** | **0.889** |

**Weighting Strategy for Hyperplane** We demonstrate the effectiveness of our weighting strategy, described in Eq. (4), by comparing human alignment score and ground truth test accuracy in Tab. 4a. The weighting strategy enables `AugCLIP` to prioritize the integration of key features that are central to preservation and modification, into the representation of an ideally edited image.

**Choice of Linear Hyperplane** We compare latent discriminant analysis, linear SVM, and logistic regression to evaluate their effectiveness in finding the separating hyperplane. As shown in Tab. 4b, linear SVM yields minimum misclassification over all benchmark dataset, in which the hyperplane sufficiently divide source attributes and target attributes. Since source image and target text may entail some visual similarities, the extracted source and target descriptions cannot be perfectly separable by a linear hyperplane. Therefore, the usage of SVM, that can flexible manage overlapping factors and find a more accurate hyperplane, leads to better performance over other hyperplanes. For instance, when editing an image of an orange to resemble a tangerine, both source and edited images have a round shape. In such cases, these factors are closely positioned in the embedding space and do not need to be perfectly separated.

**Length and Number of Descriptions**  Since `AugCLIP` employs descriptions extracted by LLM, therefore we analyze the impact of variation in descriptions on evaluation results. In Tab. 4c, we compare the cases where the attribute extraction process is restricted by description length and the total number of descriptions. Among short and long descriptions, we observe that short descriptions tend to outperform long descriptions over five benchmark datasets. This explains that short descriptions correspond to a single visual attribute, preventing the unwanted entanglement of multiple attributes into a single description. The number of descriptions is configured among 10, 20, and 30. The length of descriptions showed varying performance depending on the dataset type. Unrestricting the number of descriptions achieves the best overall performance over all configurations.

## 5.3 Application of `AugCLIP` in Diverse Editing Scenarios

Text-guided image editing spans a wide range of tasks, including style editing, object replacement, partial alteration, texture change, color change, and shape change. Given the diverse editing scenarios covered in the EditVal dataset, we report the human alignment score, $s_{align}$, for each specific editing scenario to demonstrate the effectiveness of our metric, `AugCLIP`, in handling various types of text-guided image editing tasks. Over all of the eight scenarios, `AugCLIP` outperforms CLIPScore, except for the texture modification task.

**Table 5: Human Alignment Score $s_{\mathbf{align}}$ on Various Text-guided Image Editing Scenarios** We report the alignment score of CLIPScore and `AugCLIP` over eight variants of text-guided image editing tasks in EditVal. Best scores are emphasized in bold.
*Pos. Add and Obj. repl. denotes positional addition and object replacement respectively

|  | Pos. Add | Obj. repl. | Alter Parts | Background | Texture | Color | Action | Style |
|---|---|---|---|---|---|---|---|---|
| **CLIPScore** | 0.667 | 0.688 | 0.730 | 0.5 | **0.806** | 1.0 | 1.0 | 0.529 |
| `AugCLIP` | **1.0** | **0.75** | **0.838** | **1.0** | 0.742 | **1.0** | **1.0** | **0.647** |

## Limitations

While `AugCLIP` demonstrates strong performance in balancing preservation and modification in text-guided image editing, several limitations remain. First, the reliance on GPT-4V for visual attribute extraction can lead to inconsistencies, especially in complex scenarios where subtle details are crucial. The quality of extracted attributes may vary depending on the specificity of the scene and the quality of the model's understanding, which can affect the accuracy of the modification vector. Additionally, `AugCLIP` requires longer computation times due to the need for detailed description generation and the optimization process involved in fitting the hyperplane. This makes it less efficient for real-time or large-scale applications where rapid evaluation is necessary.

## Conclusion

In this paper, we introduce AugCLIP, a novel evaluation metric for text-guided image editing that balances both preservation of the source image and modification toward the target text. By leveraging a multi-modal large language model to extract fine-grained visual attributes and applying a hyperplane-based optimization approach, AugCLIP estimates a representation of a well-edited image that closely aligns with human evaluators' preferences. Extensive experiments across five benchmark datasets demonstrate AugCLIP's superior alignment with human judgments compared to existing metrics, particularly in challenging editing tasks. Consequently, AugCLIP offers a significant advancement in the evaluation of text-guided image editing, providing a more nuanced and reliable approach for assessing modifications while maintaining core image attributes. This metric holds promise for broader applications in personalized image editing and other vision-language tasks.

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
