## A    QUALITATIVE RESULTS

We present qualitative samples of the **2AFC Test**, as reported in Tab. 2a, using the CelebA, EditVal, and DreamBooth datasets. For each dataset, we randomly selected triplets consisting of a source image, target text, and edited images to demonstrate how `AugCLIP` consistently assigns higher scores to the edited image preferred by human evaluators. The preferred image, highlighted with a red box, appears in the middle. Each case represents a two-alternative forced choice (2AFC) survey, where the source image on the far left is altered into the middle and rightmost images. We observe that CLIPScore often favors excessively modified images. For instance, in Fig. 4, where the target text is "high arch of the eyebrows," CLIPScore prefers an edited image that changes the gender of the source image into a man. Similarly, when the target text is "wrinkle-free skin," CLIPScore assigns a higher score to an image where the hair bangs are missing. This pattern is consistently observed across all three datasets, as shown in Fig. 4, Fig. 5, and Fig. 6.

Additionally, we provide qualitative samples from the **Ground Truth Test**, reported in Tab. 2b, using the TEdBench and MagicBrush datasets (Fig. 7 and Fig. 8). In these cases, the ground truth image is located in the second column, the excessively preserved image in the third column, and the excessively modified image in the fourth column. Once again, we observe that CLIPScore tends to prefer excessive modifications.

### A.1    CELEBA

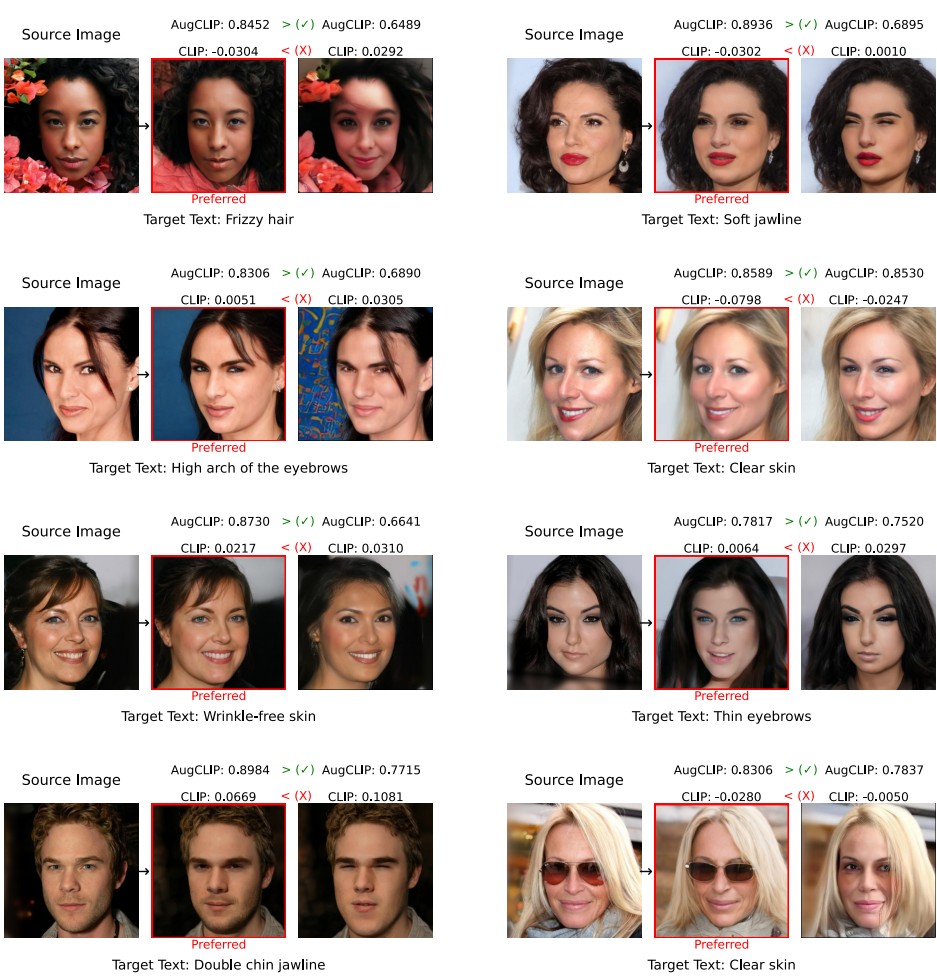

**Figure 4: Qualitative Results on CelebA (2AFC Test)**.

## A.2    EDITVAL

**Figure 5: Qualitative Results on EditVal (2AFC Test).**

## A.3 DREAMBOOTH

**Figure 6: Qualitative results on DreamBooth dataset (2AFC test).**

## A.4 TEdBench

Figure 7: Qualitative Results on TEdBench (Ground Truth Test).

## A.5 MagicBrush

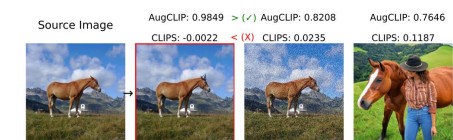

Target Text: A hotel looking room with a white curtain and bright striped bedspread.

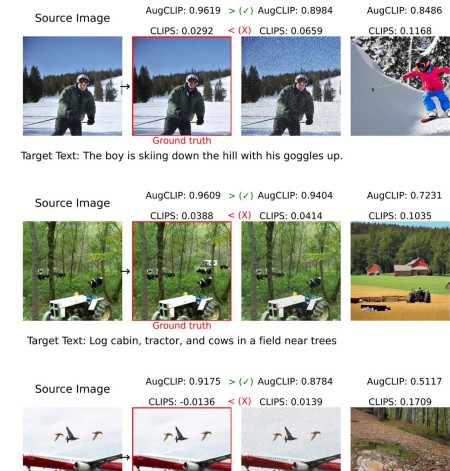

Target Text: The boy is skiing down the hill with his goggles up.

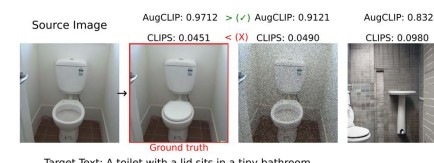

Target Text: A brown horse wearing a hat standing on top of a lush green hillside.

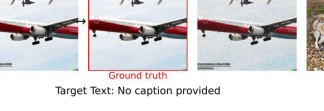

Target Text: Log cabin, tractor, and cows in a field near trees

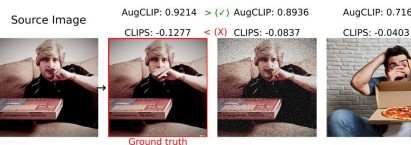

Target Text: A toilet with a lid sits in a tiny bathroom.

Target Text: No caption provided

Target Text: Man on couch with a box of pizza covering his mouth with his hand

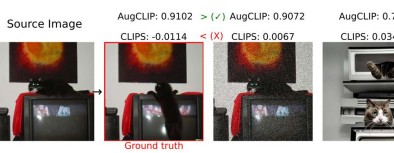

Target Text: Cat hanging from two stacked televisions.

**Figure 8: Qualitative Results on MagicBrush (Ground Truth Test).**

## B    EXPERIMENTAL DETAILS

### B.1    ASSETS

**Table 6: Assets Employed in Our Experiments**. List of pre-trained models, benchmark datasets, and metrics employed in this paper.

| Category | Asset | URL |
|---|---|---|
| **Benchmarks** | CelebA (Liu et al., 2015) | https://mmlab.ie.cuhk.edu.hk/projects/CelebA.html |
| | TedBench (Kawar et al., 2022) | https://github.com/imagic-editing/imagic-editing.github.io/tree/main/tedbench |
| | EditVal (Basu et al., 2023) | https://github.com/deep-ml-research/editval_code |
| | DreamBooth (Ruiz et al., 2023) | https://github.com/google/dreambooth |
| | MagicBrush (Zhang et al., 2024) | https://github.com/OSU-NLP-Group/MagicBrush |
| **Editing Models** | Imagic (Kawar et al., 2022) | https://github.com/huggingface/diffusers/tree/main/examples/community#imagic-stable-diffusion |
| | InstructPix2Pix (Brooks et al., 2022) | https://github.com/timothybrooks/instruct-pix2pix |
| | DiffusionCLIP (Kim & Ye, 2021) | https://github.com/gwang-kim/DiffusionCLIP |
| | DDS (Hertz et al., 2023) | https://github.com/google/prompt-to-prompt/blob/main/DDS_zeroshot.ipynb |
| | Plug-and-Play (Tumanyan et al., 2023) | https://github.com/MichalGeyer/plug-and-play.git |
| | DiffEdit (Couairon et al., 2022) | https://github.com/Xiang-cd/DiffEdit-stable-diffusion.git |
| | Prompt-to-Prompt (Hertz et al., 2022) | https://github.com/google/prompt-to-prompt.git |
| | MasaCtrl (Cao et al., 2023) | https://github.com/TencentARC/MasaCtrl.git |
| | Text2Live (Bar-Tal et al., 2022) | https://github.com/omerbt/Text2LIVE |
| | StyleCLIP (Patashnik et al., 2021) | https://github.com/orpatashnik/StyleCLIP |
| | Multi2One (Kim et al., 2022) | https://github.com/akatigre/multi2one |
| | Asyrp (Kwon et al., 2022) | https://github.com/kwonminki/Asyrp_official |
| **Metrics** | CLIP (Radford et al., 2021) | https://github.com/openai/CLIP |
| | CLIPScore (Gal et al., 2022) | Implemented by Authors |
| | LPIPS (Zhang et al., 2018) | https://pypi.org/project/lpips/ |

### B.2    DESCRIPTION GENERATION PROCESS

We leverage GPT-4V (OpenAI, 2023) to extract visual attributes of the source image and target text. These attributes are presented as textual descriptions, highlighting various visual features like shape, color, texture, patterns, posture, action, and position. The number of extracted descriptions is determined by the ability of GPT-4V depending on the complexity of editing scenarios. For complex scenes, GPT-4V typically produces around 30 descriptions, while simpler scenarios, involving only a single object and basic modifications, generate roughly 5 descriptions—sufficient to capture the entire scene and intended edits. Fig. 9 shows the prompt used for attribute extraction, where example outputs ensure that each description represents a distinct visual element.

## C    DETAILS ON `AugCLIP`

### C.1    QUALITATIVE ANALYSIS ON THE EFFECT OF MODIFICATION VECTOR **v**

We modify the source image to reflect minimum modification in the source image with **v**. Thus the source attributes that should be preserved must not be harmed by adding **v** while attributes that require modification into target text must be altered. In order to show that **v** drives large change on source attributes that must be altered, and inflict small change on source attributes that must be preserved, we analyze several cases in TedBench and EditVal in Fig. 10. The difference in cosine similarity between the source image and the estimate of an ideal edited image is measured with the source attributes $s$ as $I_{\text{src}} + \mathbf{v}$ is measured as $\text{cs}(I_{\text{src}} + \mathbf{v}, s_i) - \text{cs}(I_{\text{src}}, s_i)$.

### C.2    COMPUTATION TIME

Our method, `AugCLIP`, requires extraction of descriptions via LLMs, then fitting the hyperplane between source and target attributes to derive the ideal representation. Compared to CLIPScore that simply requires similarity measurement between the image and text, our method requires 12.3 seconds for description generation, followed by 0.15 seconds for score computation. This pose extra computation time of for description generation, but description set for established benchmark dataset could be preprocessed to be reused in evaluation process, making the computation time on par with CLIPScore.

## Prompt for generating a detailed caption of the source image

[User] Describe the image in detail. Do not describe the background or opinions. Make the descriptions easy and intuitive.
Image: source_image

## Prompt for parsing generated caption to source attributes

[System] You are a helpful text generation assistant. Given a detailed textual description of an image, your goal is to parse it into specific visual attributes. If the visual attributes with similar meanings are repeated, only use one. For example, 'A dog is large' and 'A dog is big' are similar, so only use one.

[User] Description: 'The image features a large, big, black dark-colored dog standing in a grassy field. The dog appears to be alert and attentive, possibly observing its surroundings. The grass is lush and green, providing a natural backdrop for the dog.'

[Assistant] 'A dog is large', 'A dog is dark-colored', 'A dog is standing', 'A dog in standing on a grassy field', 'A dog is alert', 'A dog is attentive', 'A dog is observing its surroundings', 'A grass is lush', 'A grass is green', 'A grass is providing a natural backdrop'

[User] Description: generated_caption

## Prompt for generating target attributes

[System] You are a helpful text generation assistant. Given a textual description, your goal is to list specific visual attributes.

[User] In order to make a sitting person into a standing person, what visual attributes of the image should be changed? Answer in the format of 'A standing person is ATTRIBUTE'.

[Assistant] 'Standing person has straight legs', 'Standing person is upright', 'Standing person is on their feet', 'Standing person is tall'

[User] In order to make source_text into target_text, what visual attributes of the image should be changed? Answer in the format of 'target_text is ATTRIBUTE'.

**Figure 9: Prompt for Visual Attribute Extraction.**

## C.3 Benchmark Datasets for Text-guided Image Editing

TEdBench comprises 100 pairs of source image and target text. It focuses on specific settings where the source image has a single object at the center, and the corresponding target text only modifies some attributes of that object.

EditVal contains 648 image-text pairs that cover 13 different types of edits, including object addition, object replacement, and size modification. Since it has such complicated editing scenarios, models that we leveraged could not properly edit the most cases so that there are not much samples with enough quality for user study. So, we use eight edit types for evalaution.

MagicBrush is a benchmark specifically designed to evaluate sequential editing tasks, where iterative modifications are made to different parts of the source image. Dreambooth enables the modification of specific instances within the source image by providing corresponding masks along with image-text pairs; however, since typical editing models do not utilize masks as input, we only consider the image-text pairs in our evaluation.

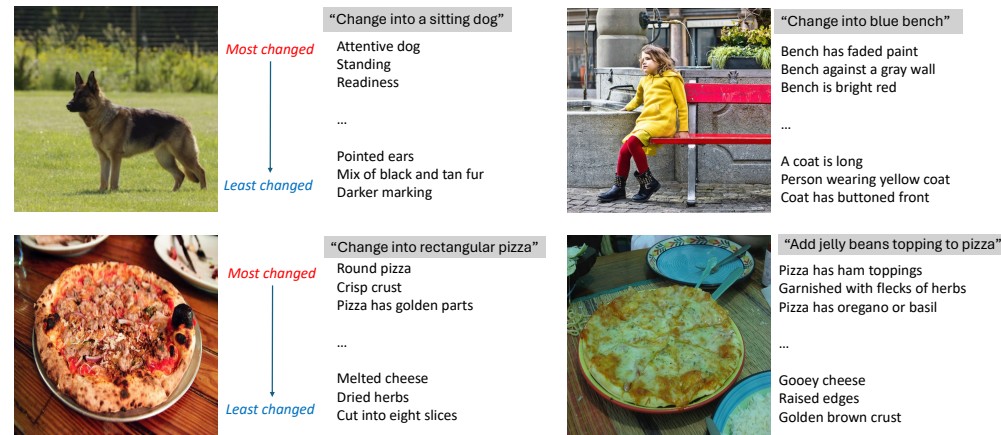

**Figure 10: Effect of v in Source Attributes** The source descriptions are listed in the order of largest alteration to smallest alteration caused by adding the modification vector $s_{\text{proj}}$. The text descriptions listed on the top signify source attributes that must be modified towards the target text.

Finally, for the CelebA dataset, we create a subset consisting of 50 image-text pairs that guide changes specific to facial attributes. We created the prompt by swapping attributes of human face.

### C.4 Comparison with GPT-4V

**Table 7: Comparison with GPT-4V.** We use 2AFC scores for CelebA, EditVal, and Dreambooth, and $\textbf{Acc}_{\text{both}}$ for TEdBench and MagicBrush.

|          | CelebA | EditVal | DreamBooth | TEdBench | MagicBrush |
|----------|--------|---------|------------|----------|------------|
| GPT-4V   | 0.876  | **0.933** | 0.821      | **0.620** | 0.703      |
| AugCLIP  | **0.883** | 0.831 | **0.857**  | 0.570    | **0.889**  |

Recently, GPT-4V has been employed in evaluating various vision-language tasks, including text-guided image editing, text-to-image generation, and image quality assessment. In this study, we analyze GPT-4V's effectiveness in evaluating the quality of text-guided edited images, focusing on both preservation and modification aspects. As shown in Tab. 7, GPT-4V outperforms AugCLIP in tasks such as EditVal and TEdBench, which involve simple edits like modifying a single object's attribute. This finding is consistent with prior research (Zhang et al., 2023), which suggests that GPT-4V struggles to differentiate between images with subtle differences. In contrast, our proposed metric, AugCLIP, effectively captures minor differences by augmenting attributes of the source image and target text and shows better performance in other benchmarks.

### D Existing evaluation metrics

FID (Heusel et al., 2017) and IS (Salimans et al., 2016) evaluate the diversity and quality of generated images by analyzing the output of a pre-trained classifier. They only assess the fidelity of the edited image, regardless of the model inputs.

LPIPS (Zhang et al., 2018), DINO similarity(Caron et al., 2021) and Segmentation Consistency (Kim & Ye, 2021) evaluate the preservation of source image in terms of distributional change in extracted feature and change in segmentation maps. These metrics do not consider how the source image should be modified accordingly with the given target text.

Several metrics evaluate the alignment between the edited image and the text guidance (Hessel et al., 2021), relying on vision-language models (Radford et al., 2021; Minderer et al., 2022).

## D.1 Combination of Preservation and Modification Centric Metrics

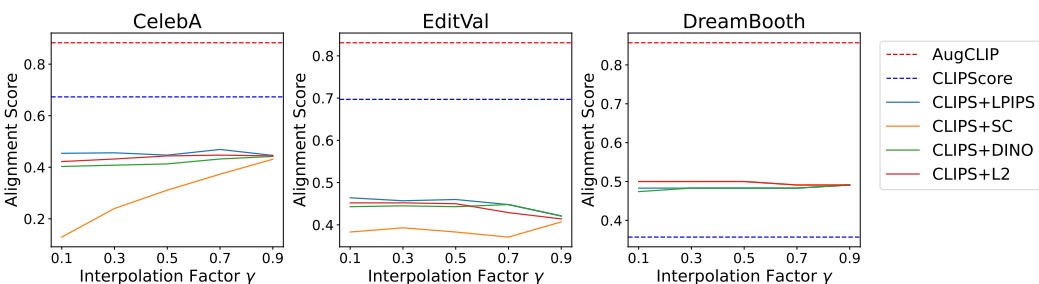

**Figure 11: Combination of Preservation-centric Metrics with CLIPScore** The plot shows the human alignment score $s_{\mathrm{align}}$ over three benchmark dataset, CelebA, EditVal, and DreamBooth, when compared with a linear interpolation of preservation centric metrics with CLIPScore. The result shows that even with combination of preservation score, CLIPScore fails to align with human judgment.

We explore if combining preservation and modification metrics could lead to improvement with human judgment with three datasets, CelebA, EditVal and DreamBooth. We combine the two metrics, CLIPScore and one of the preservation metrics among LPIPS, Segment Consistency, DINO similarity and L2 with interpolation value of $\gamma$. Specifically, the scores are computed as CLIPScore $\times \gamma$ + Preservation score $\times (1-\gamma)$. In CelebA and EditVal, combination negatively affect the alignment with human evaluation, as using CLIPScore alone leads to much higher alignment. In DreamBooth, the combinations outperforms CLIPScore but falls short of our metric `AugCLIP` by a large margin. Note that the two scores are scaled into the same range before linear interpolation, to ensure that intended proportion of $\gamma$ is integrated into the final combined score.

# E    USER STUDY DETAILS

## Instruction for user study

This user study is part of a research project on evaluating text-guided image manipulation.

SITUATION)
Each sample will display a text prompt at the top, an original image on the left, and two manipulated images on the right.

CRITERIA FOR GOOD MANIPULATION)
1. Realism: The manipulated image should possess high realism, aiming to appear as authentic as possible.
2. Relevance to Text Prompt: The manipulated image should be "closely aligned with the accompanying text prompt" while "preserving the original image's essence". For instance, if the text prompt is "Change a dog to a cat," the color and posture of the dog in the original image and the cat in the manipulated image should correspond.

Your meticulous assessment of the images on each page is greatly appreciated, as it contributes significantly to the success of our research. Thank you!

## Example format of the user study question

Source text: a backpack
Target text: a backpack in the jungle

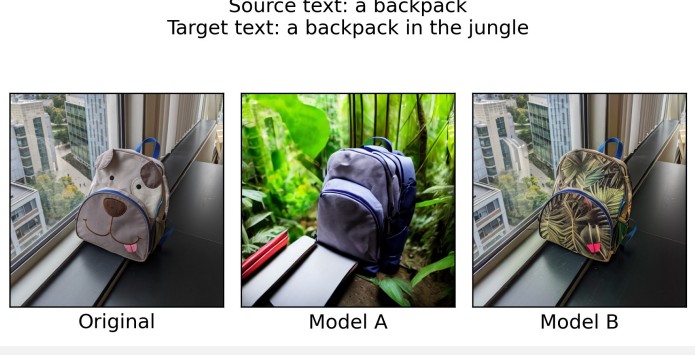

Original            Model A            Model B

**Figure 12: User Study Examples**.

**Table 8: User Study Statistics for Different Datasets**.

|  | CelebA | EditVal | DreamBooth |
|---|---|---|---|
| **Survey questions** | 39 | 35 | 37 |
| **Total image-text pairs** | 50 | 648 | 3950 |
| **Participants** | 45 | 30 | 30 |

Due to the limitations of existing text-guided image editing models, which often struggle to produce high-quality edited images, we manually selected samples of sufficient quality for the user study. Each participant was shown a source image, two edited images, and the target text, and asked to choose the better-edited image. As shown in Fig. 12, we provided clear guidelines, instructing participants to evaluate the images based on both the preservation of the source image and the modifications toward the target text. The survey was conducted using Amazon Mechanical Turk, allowing us to gather responses from a diverse group of participants. Details on the user study and datasets are available in Tab. 8.