# OpenReview forum: "Augmentation-Driven Metric for Balancing Preservation and Modification in Text-Guided Image Editing"
_ICLR.cc/2025/Conference — ICLR 2025 Conference Withdrawn Submission_

### Official Review · Reviewer_QH3u · 2024-10-30

**Soundness:** 3
**Presentation:** 3
**Contribution:** 3
**Rating:** 6
**Confidence:** 4

**Summary:**

This paper introduces AugCLIP, a novel evaluation metric for text-guided image editing that balances both preservation of the source image and modification toward the target text. By leveraging a multi-modal large language model to extract fine-grained visual attributes and applying a hyperplane-based optimization approach, AugCLIP estimates a representation of a well-edited image that closely aligns with human evaluators’ preferences. Extensive experiments across five benchmark datasets demonstrate AugCLIP’s superior alignment with human judgments compared to existing metrics, particularly in challenging editing tasks. Consequently, AugCLIP offers a significant advancement in the evaluation of textguided image editing, providing a more nuanced and reliable approach for assessing modifications while maintaining core image attributes. This metric holds promise for broader applications in personalized image editing and other vision-language tasks.

**Strengths:**

The paper introduces a novel evaluation metric for text-guided image editing that balances both preservation of the source image and modification toward the target text. It demonstrates remarkable improvement in alignment with human evaluators on diverse editing scenarios such as object, attribute, style alteration compared to all other existing metrics. Moreover, the metric is applicable to personalized generation, DreamBooth dataset, where the objective is to identify the source object in provided image, and generate into a completely novel context. This shows the flexibility of AugCLIP, that seamlessly apply to variety of editing directions. Notably, the metric excels in identifying minor differences between the source image and the edited image, showing superb ability in complex image editing scenarios such as MagicBrush. The major contributions are summarized as follows.
- This paper is the first to point out CLIPScore’s reliability in text-guided image editing, as it frequently exhibits a bias towards modification rather than preservation and focuses on irrelevant regions.

- This work proposes AugCLIP, a metric for image editing by automatically augmenting descriptions via LLM and estimating a balanced representation of preservation and modification, which takes into account the relative importance of each description.

- In the experimental  evaluations, AugCLIP demonstrates a significantly high correlation with human evaluations across various editing scenarios, even in complex applications where existing metrics struggle.

**Weaknesses:**

Overall this work makes an interesting and meaningful observation about the widely used CLIPScore metric, however there are still some concerns:
- **Discussion on broader indicators.** This work highlights the problem of CLIPScore in the problem analysis in Section 3. Do other quantitative indicators such as FID and LPIPS have similar problems? Please give a more comprehensive analysis.
- **Suitability for complex editing instructions or tasks.**  There are many kinds of image editing tasks, including global editing such as style editing rather than just local editing. How does AugCLIP perform in this case?

**Questions:**

See weaknesses.

---

> ### Author Response · Authors · 2024-11-15
>
> Thank you very much for your thoughtful and detailed reviews. We appreciate your insights and would like to address the points raised:
>
> 1. FID and LPIPS are metrics that do not take the target text into account at all, meaning they cannot evaluate whether an image has been modified in accordance with the target text. The reason we focused on analyzing CLIPScore (directional CLIP similarity) is that it is the only metric that jointly considers the source image and the target text.
>
> 2. As demonstrated in Section 5.3, AugCLIP is adaptable to various editing scenarios. Furthermore, the benchmark datasets we used in the paper, as described in Appendix C.3 and detailed in Appendix A, encompass a wide range of contexts. This versatility allows for fine-grained evaluation of complex editing tasks through the use of descriptions generated by MLLMs.
>
> We are grateful for your valuable feedback and will make the necessary revisions to enhance the clarity and quality of our paper. Thank you once again for your time and insights.

---

### Official Review · Reviewer_zaUq · 2024-11-02

**Soundness:** 2
**Presentation:** 2
**Contribution:** 2
**Rating:** 3
**Confidence:** 4

**Summary:**

This paper introduces AugCLIP, an evaluation metric designed for text-to-image editing tasks. AugCLIP aims to address limitations in CLIPScore, which can not evaluate the preservation of the original input image. The method leverages GPT-4V for detailed descriptions of the source and target images. By creating a "modification vector" based on source and target attributes, AugCLIP balances preservation and modification. The authors demonstrate that AugCLIP outperforms metrics such as LPIPS, CLIPScore on various datasets.

**Strengths:**

- The authors evaluate AugCLIP on multiple benchmarks, demonstrating that AugCLIP outperforms CLIPScore and LPIPS with various editing methods and datasets.

- AugCLIP can evaluate both the modification and preservation of the editing images. Compared to CLIPScore, AugCLIP is a more comprehensive metric.

- By leveraging GPT-4V, AugCLIP can evaluate more fine-grained differences between the ground truth image and the edited image.

- The authors provide ablation studies to evaluate difference components of AugCLIP.

**Weaknesses:**

- It seems that the authors overclaimed their contributions. For example, in Line 081-083, they mentioned that "We are the first to point out CLIPScore’s reliability in text-guided image editing". As far as I know, many papers have pointed out the limitations of CLIPScore. Almost all image editing methods leverage CLIP to evaluate the modification, and LPIPS/FID to evaluate the preservation. For example, [1,2] provides both CLIP and LPIPS to evaluate the editability–fidelity tradeoff.

- The authors seem to confuse CLIP score with CLIP directional similarity score (*i.e.*, directional CLIP loss). From my understanding, the definition in Section 3.1 is more like CLIP directional similarity score rather than CLIP score. Please double check the definition of CLIP and CLIP similarity score in the following link:
 https://huggingface.co/docs/diffusers/conceptual/evaluation.

- The experiments only involves metrics like LPIPS, CLIP. Please consider include the tradeoff between CLIP and 1-PIPS or FID.

- Introducing GPT-4V introduces additional overhead, which is not evaluated in the related experiments.

[1] Zhang, Zhixing, et al. "Sine: Single image editing with text-to-image diffusion models." CVPR 2023.
[2] Kawar, Bahjat, et al. "Imagic: Text-based real image editing with diffusion models." CVPR 2023.

**Questions:**

Please refer to **Weaknesses**.

---

> ### Author Response · Authors · 2024-11-15
>
> Thank you very much for your thoughtful and detailed reviews. We appreciate your insights and would like to address the points raised:
>
> $\textbf{Overclaim of contribution}$ We have not overclaimed the contribution, since we first analyzed the shortcomings of “directional CLIP similarity” specifically in the context of “text-guided image editing”, not the shortcomings of "CLIP-score". As the terminology is confusing, we will revise the paper.
>
> $\textbf{Experiments involve all suggested variants}$ Combination of existing metrics have been demonstrated in Appendix D.1. FID is not measured in a sample-wise manner, which is impossible be applied in our experiments. I hope this clarifies our experiments.
>
> $\textbf{Addition overhead is already analyzed}$ Please refer to Appendix C.2. for additional computational overhead.
>
> We are grateful for your valuable feedback and will make the necessary revisions to enhance the clarity and quality of our paper. Thank you once again for your time and insights.

---

### Official Review · Reviewer_LwcK · 2024-11-02

**Soundness:** 3
**Presentation:** 3
**Contribution:** 2
**Rating:** 5
**Confidence:** 4

**Summary:**

This paper proposed a novel evaluation metric, augclip,  for text-guided image editing. Motivated by the observation that clip score biases towards modification instead of preservation, the authors utilized GPT-4v to rephrase  the source prompt and target prompt to extract essential visual attributes in the form of text prompts.  Then the authors trained classification models to classify source prompts from target prompts. Since CLIP itself aligns the image space and text space, the classification model trained with source prompts and target promtps could also be utilized to  compute the minimum vector  v that transfer source image embedding to target text embedding.  The augclip metric is then defined to be the cosine similarity of image embedding of the edited image and the vector sum of  v and source image embedding. The proposed augclip demonstrate superior alignment with human evalutions than clip score and lpips score.

**Strengths:**

1. The observation  and visualization that clip score favors modifications instead of preservation is interesting.

2.  The idea to transfer the trained SVM from text space to image space and  compute the minimum v from source image embedding to target prompt embedding is clever.

3.  the two-alternative forced choice testing and ground truth testing are reasonable.

**Weaknesses:**

1. Since the main contribution of this paper is the augclip evaluation metric, the authors should compare with more than one image editing method on each benchmark. However, the authors only evaluated the results of one image editing method, the results of stable diffusion 1.5 (which is just a text to image base model without editing capability itself) and the original reference image,  with the proposed augclip. For a new evaluation metric, this is far from enough. For example, the authors showed the reference images from TEdBench [2] multiple times in the paper, yet they only evaluate the scores of Imagic+Imagen. There are other related works on this benchmark, for example, Forgedit[3] open-sourced their implementation and released their results on TEdBench on github.

2.  Incorrect clip score definition. The clip score in this paper, shown in equation 1 in section 3.1, is different from  the usual clipscore being used in text-guided image editing literature [1].  For example, in Imagic[2] and Forgedit[3], the clip score metric's definition follows [1].

3. Most editing methods in table 6 in  the appendix  never appear in the paper and section 4.3 is not written well thus is very confusing.




[1] Clipscore: A referencefree evaluation metric for image captioning. In EMNLP

[2] Imagic: Text-based real image editing with diffusion models. In CVPR

[3] Forgedit: Text-guided Image Editing via Learning and Forgetting. In arxiv

**Questions:**

1. The main contribution of this paper is the new metric,  augclip, for text-guided image editing. However, very few editing methods are tested with this new metric, which weakens the solidness of the metric. Considering the limited rebuttal period, I will not ask the authors to evaluate all editing methods in table 6 in the appendix. However, since the main benchmark illustrated in the paper is TEdBench[2], I suggest the authors to evaluate Forgedit[3]  with augclip  since they also released the complete editing results of TEdBench on github.  You have to compare image editing methods with augclip to demonstrate its effectiveness instead of text-to-image models like stable diffusion itself in your paper.

2.  The clip score definition in equation 1 is different from the mainstream reference[1]. Where does this equation 1 come from? Why is it used instead of [1]?

3. How long does it take to train the augclip metric on each benchmark?


I am willing to raise my rating score if the authors could stress my concerns in the revised version of this paper.

references:

[1] Clipscore: A referencefree evaluation metric for image captioning. In EMNLP

[2] Imagic: Text-based real image editing with diffusion models. In CVPR

[3] Forgedit: Text-guided Image Editing via Learning and Forgetting. In arxiv

---

> ### Author Response · Authors · 2024-11-15
>
> Thank you very much for your thoughtful and detailed reviews. We appreciate your insights and would like to address the points raised:
>
> 1. In the ground truth test, we intentionally generated excessively preserved or modified samples using SD-1.5. However, for the Two-Alternative Forced Choice (2AFC) test, we manipulated images using the editing methods listed in Table 6 and created the survey based on these manipulated images. We will revise Section 4.3 to clarify this explanation further.
>
> 2. The CLIPScore we define in our paper corresponds to directional CLIP similarity and is the only metric leveraging CLIP within the context of text-guided image editing. We acknowledge that this may have caused confusion, and we will make the necessary revisions to ensure clarity.
>
> 3. Please refer to our response to point 1 for related details.
>
> 4. Regarding Question 1, thank you for suggesting an excellent experimental idea. We will incorporate your proposed approach in future experiments.
>
> 5. For Question 2, please refer to our response to point 2 for further clarification.
>
> 6. For Question 3, details related to computation time can be found in Appendix C.2.
>
> We are grateful for your valuable feedback and will make the necessary revisions to enhance the clarity and quality of our paper. Thank you once again for your time and insights.

---

### Official Review · Reviewer_dg3W · 2024-11-03

**Soundness:** 3
**Presentation:** 3
**Contribution:** 2
**Rating:** 3
**Confidence:** 3

**Summary:**

In this paper, the author proposes a new metric called AugClip, for Text-guided Image Editing.

**Strengths:**

The author shows the disadvantage of ClipScore, and try to design a new one.

1. The key question is that we can use fusion of other existing metric without introducing a new one.
2. The formulation of clipscore is not common.

**Weaknesses:**

1. Some notations are confusing.
T sometimes is text, while T sometimes is the set.

2. Overclaims in contribution-1 "We are the first to point out CLIPScore’s reliability in text-guided image editing"
In fact, most researchers recognize this point, and use a cocktail metric, like FID + Clipscore + SSIM + human evaluation.
[a] Holistic Evaluation of Text-to-Image Models. NeurIPS.

3. Conflicts in contribution-2
In abstract, the author said using MLLM.
In introduction, the author claims LLM.

4. Contribution-3 said "demonstrates" but there is no mathmatic proof.

5. Figure 1 does not convince me.
We could simply use cosine(f_source, f_edit) to see the preservation.
https://openaccess.thecvf.com/content/WACV2024/papers/Tanjim_Discovering_and_Mitigating_Biases_in_CLIP-Based_Image_Editing_WACV_2024_paper.pdf

6. Figure 2 is similar to Figure 1.
We could simply use cosine(f_source, f_edit) to see the preservation.

7. Eq.1 is not commonly-used.
Could you show the reference? It does not make sense, since clip feature can not use plus or minus operation.
Most cases I read is using cosine(f_modification text, f_editted image)

8. One simple ablation is missing.
How about the weighted sum like cosine(f_modification text, f_editted image) + 0.5*cosine(f_source image, f_editted image) ?
cosine(f_modification text, f_editted image)  higher is better modification.
cosine(f_source image, f_editted image)  higher is better preservation.
Usually, we will use the FID to indicate the preservation as well.

9. I am confusing about Eq.3,4,5.
Eq 4,5 is about a , not v.
Eq 3 is about v, not a.
But the author said a can control v.  "the refined version of v is obtained through hyperplane optimization using as and at."

**Questions:**

Please see the weakness.

1. The key question is that we can use fusion of other existing metric without introducing a new one.
2. The formulation of clipscore is not common.

---

> ### Author Response · Authors · 2024-11-15
>
> Thank you very much for your thoughtful and detailed reviews. We appreciate your insights and would like to address the points raised:
> $\textbf{Overclaim of contribution}$ We have not overclaimed the contribution, since we first analyzed the shortcomings of “directional CLIP similarity” specifically in the context of “text-guided image editing”, not the shortcomings of "CLIP-score". As the terminology is confusing, we will revise the paper.
>
> $\textbf{Usage of cocktail metrics are already shown to be ineffective in the paper}$ Combination of existing metrics have been demonstrated in Appendix D.1. FID is not measured in a sample-wise manner, which is impossible be applied in our experiments. I hope this clarifies our experiments.
>
> $\textbf{Misunderstanding in Figure 1, 2}$ Figure 1 is to show excessive modification, not preservation. Also, Figure 2 shows how directional CLIP similarity attend to edited regions, which is irrelevant with calculation of cosine(f_source, f_edit).  We hope this clarifies our intention.
>
> $\textbf{Weighting strategy}$ The weights derived from Eq. 4 and 5 are applied as weightings for each attribute in the hyperplane optimization process. We will revise the paper to explicitly explain this procedure.
>
> We are grateful for your valuable feedback and will make the necessary revisions to enhance the clarity and quality of our paper. Thank you once again for your time and insights.

---

### Note · Authors · 2024-11-15

**Comment:**

Thank you very much for your valuable reviews. We acknowledge that the mixed usage of CLIPScore and Directional CLIP similarity may have caused some confusion, and we plan to address this issue for better clarity. Additionally, we will make sure to explicitly define the scope of our comparisons, particularly in the context of text-guided image editing, to avoid any ambiguity. Your feedback is greatly appreciated, and we will strive to improve our work accordingly.

**Withdrawal Confirmation:**

I have read and agree with the venue's withdrawal policy on behalf of myself and my co-authors.